# Self-Sustainable Biomedical Devices Powered by RF Energy: A Review

**DOI:** 10.3390/s22176371

**Published:** 2022-08-24

**Authors:** Hussein Yahya Alkhalaf, Mohd Yazed Ahmad, Harikrishnan Ramiah

**Affiliations:** 1Department of Biomedical Engineering, Universiti Malaya, Kuala Lumpur 50603, Malaysia; 2Department of Electrical Engineering, Universiti Malaya, Kuala Lumpur 50603, Malaysia

**Keywords:** implantable medical devices, rectenna, RF energy harvesting, wearable medical devices, wireless power transfer

## Abstract

Wearable and implantable medical devices (IMDs) have come a long way in the past few decades and have contributed to the development of many personalized health monitoring and therapeutic applications. Sustaining these devices with reliable and long-term power supply is still an ongoing challenge. This review discusses the challenges and milestones in energizing wearable and IMDs using the RF energy harvesting (RFEH) technique. The review highlights the main integrating frontend blocks such as the wearable and implantable antenna design, matching network, and rectifier topologies. The advantages and bottlenecks of adopting RFEH technology in wearable and IMDs are reviewed, along with the system elements and characteristics that enable these devices to operate in an optimized manner. The applications of RFEH in wearable and IMDs medical devices are elaborated in the final section of this review. This article summarizes the recent developments in RFEH, highlights the gaps, and explores future research opportunities.

## 1. Introduction

Biomedical devices (wearable and implantable) have expanded in popularity in integrating digital health for a diversity of biomedical applications, which includes the sensing, tracking, and monitoring of vital signals for continuous monitoring, thus making it possible for early intervention. In the healthcare sector, the advancement in these technologies is a result of several factors such as the increasing population of people with chronic diseases (such as diabetes and obesity), the aging population, and the rising need for real-time health monitoring, including fitness and wellness [1]. Recent developments in integrated circuits, medical sensors, wireless communications, and fabrication methods enable small, low-power devices with special interfacing capabilities to interact with human tissues and biological objects [2]. The platforms offered by these interfaces are appropriate for the quantitative measurement, continuous observation, and documentation of physiological and biomedical parameters, as well as for the modification of cells, tissues, or organs. Figure 1 shows different biomedical devices used in various body locations. The devices are usually powered by conventional batteries that are either rechargeable or can be replaced periodically [3]. Battery technology has advanced and now offers compact sizes with high energy capacities. However, the functional lifespan of the batteries is still relatively short and demands frequent replacement. Additionally, it is adverse and impractical to entirely power these biomedical devices with conventional batteries, especially the stretchable devices which are too small and thin to even accommodate a battery. Another issue is that the process involved in replacing the battery of these devices may trigger unnecessary side effects such as infection, inflammation, internal bleeding, and prolonged recovery [4]. Consequently, the significant challenge of these small-scale wearable and IMDs is to have a sufficient sustainable power source that is stable, biocompatible, and with no side effects. Renewable energy sources, such as RFEH technology, have great potential to fulfill the above requirements because the ambient or near-field RF energy are ubiquitous and always available. RFEH technology is based on scavenging the electromagnetic energy from RF signals. This technology enables self-sustained or battery-assisted biomedical devices with an extended lifespan. The RF wave is available in both outdoors and indoors, in urban, suburban, and rural regions. RF sources can be categorized into the following categories: ambient sources (e.g., near cell phone towers) and dedicated/nearfield sources such as Powercast TX91501-3WID in a wireless powered communication network [5]. An ambient source emits stable output power to supply wireless sensors irrespective of the transmission frequency. On the other hand, a dedicated source provides a predefined energy supply to the device by optimizing the frequency and maximum power transfer to replenish sufficient power for the sensor devices.

The exponential growth in the broadcasting infrastructures and wireless communication with the advancement in the broadcasting technologies have led to an increase in the availability of ambient RF power density. The open-space medium is populated by sources of electromagnet signal such as wireless fidelity (Wi-Fi) signals [6,7,8,9,10,11], AM/FM radio [12,13], mobile base stations [14,15,16,17,18,19], television (TV), and digital television (DTV) [20,21,22,23], as described in Figure 2. A comparison of various power sources available for energy harvesting is shown in Table 1, considering the characteristics of these sources for energy harvesting: power density, efficiency, advantage, and bottleneck. The advantages of the RFEH technology over the other energy harvesting methods are as follows:The widespread availability of RF energy makes it suitable to supply sustainable energy for various biomedical devices when compared with other energy harvesting technologies such as thermal, solar, and vibration.Unlike solar, RFEH is a cost-effective technology and can be used to power the IMDs.In comparison with solar, kinetic, and thermal energy, RFEH does not depend on the light source, body motion, or temperature.

A typical RFEH system comprises a receiving antenna, a matching circuit, a rectifier, a power management unit (PMU), and the load, as illustrated in Figure 2. The receiving antenna captures the incident RF power, whereas the role of the rectifier circuit is to convert the received RF to DC. In addition, a matching circuit is required to maximize the received input power. PMU stocks and manages the harvested RF energy and also supplies the load.

The recently published review papers on energy harvesting for biomedical devices [24,25,26,27,28,29] investigate multiple technologies to power up these devices. To the best of our knowledge, this is the first review that extensively discusses and focuses on recent RF energy harvesting technologies to power up wearable and IMDs. Figure 3 illustrates the organization of this review, in which Section 2 presents the operation principle of the RFEH system, antenna design and characteristics, impedance matching network, and rectifier topologies. Section 3 highlights the application of RFEH in wearable and IMDs, followed by the conclusion.

**Table 1 sensors-22-06371-t001:** Characteristics of various sources for energy harvesting.

Reference	Energy source	Power density	Efficiency (%)	Advantage	Bottleneck
[30,31]	Perovskite solar cells	35.0 μW/cm${}^{2}$	25.2	Flexible and lightweight; suitable for wearable applications	Require light
[32,33,34,35]	Thermoelectric	Human: 100 μW/cm${}^{3}$ Industrial: 100 mW/cm${}^{3}$	10–15	Cost-effective technology; does not require body motion or light	Low power source
[36,37]	Acoustic	1.436 mW/cm${}^{2}$ at 123 dB	0.012	Require minimum maintenance; suitable to be used in remote or inaccessible locations	Hard to capture energy from the sounds wave source
[35,38,39,40]	Pyroelectric	3.5 μW/cm${}^{3}$ at the temperature rate of 85 °C/s @0.11 Hz	1–3.5	Cost-effective technology; ubiquitous and serves as a low-grade waste	Low output power
[41,42,43]	Piezoelectric	29.2 μW/mm${}^{3}$	83.3	Does not require RF waves or light	low power source; require body activity
[44,45,46]	Biofuel cells	3.7 mW cm${}^{-2}$	86	The integration of the power module and sensing module results in better compactness; does not require RF waves, body activity, or light	The analyte concentration influences the power density
[47,48]	Triboelectric	2.5 W/m${}^{2}$	——	Simple fabrication process and low cost	low power source; require body activity
[38]	RFEH	GSM: 0.1 μW/cm${}^{2}$ WiFi: 0.01 μW/cm${}^{3}$	50–70	Does not require light or body motion and is continuously available	Low output power; distant dependent

## 2. RF Energy Harvesting Principles

RFEH is enabled by ambient, dedicated, or unidentified RF signal sources. The amount of energy harvested using RFEH is dependent on the transmission power, the RF signal’s wavelength, and the distance between the RF power source and the harvesting node. The harvested power at the receiving antenna can be computed using the Friis equation as given in (1) for a transmitter and receiver antenna with a line of sight propagation [49].
(1)$$P_{r}=\frac{P_{t}G_{t}G_{r}\lambda^{2}}{{\left(4\pi d\right)}^{2}L}$$
where $P_{r}$ signifies received power and $P_{t}$ indicates transmitted power, $G_{t}$ denotes the transmitter of the antenna and $G_{r}$ represents receiver gain of the antenna, λ refers to transmitted wave wavelength, L denotes the path loss factor, and d defines the distance between transmission and receiving antenna.

Free space loss represents the loss in the signal strength and is computed by determining the distance between the transmitter and receiver, the transmitting frequency, and the antenna gain [50]. The equation to compute the free space path loss is given in (2–3):(2)$$PL=\frac{{\left(4\pi d\right)}^{2}}{G_{t}G_{r}\lambda^{2}}=\frac{{\left(4\pi fd\right)}^{2}}{G_{t}G_{R}c^{2}}$$
(3)$$PL\left(dB\right)=32.44+20log_{10}\left(f\right)+20log_{10}\left(d\right)-G_{t}-G_{R}$$
where PL stands for the free space path loss, c signifies the speed of light, and f denotes the frequency of the transmitted wave.

The performance parameters of an energy harvester (based on different applications requirements) are power conversion efficiency (PCE), operation distance, output power, and sensitivity. The PCE measures the rectifier efficiency to convert the RF signal to DC. Equations (4) and (5) are used to compute the PCE of a rectenna system [51]:(4)$$\eta =\frac{OutputPower}{InputPower}\times 100$$
(5)$$\eta =\frac{P_{DC}}{P_{IN}}=\frac{V_{DC}^{2}}{RL}\cdot \frac{1}{PL}$$
where $V_{\mathrm{DC}}$ indicates the *DC* output voltage, and $P_{\mathrm{IN}}$ indicates the incident power at the rectenna input.

The operational distance of a harvester is reliant on the operating frequency. The *DC* output power describes the output of the RFEH system. The output is technically described as the minimum and maximum load voltage and current of the power management system. Due to the dependency of the load voltage and the current on the load impedance, measurement in an open load condition will accurately describe the performance of the RFEH system. In some applications, such as the sensors, the load voltage is a more critical parameter than the current, but in others, such as the LED, the predominant parameter is current. For the RFEH system, the sensitivity can be described as the optimum input power required to operate. The threshold voltage of complementary metal–oxide–semiconductor (CMOS) technology impacts the sensitivity tradeoffs and in an advanced technology node, the leakage current adversely affects the efficiency.

The free space RF energy is divided into two categories: near field and far field. The frequency range of energy transfer is from 3 kHz to 3000 MHz. RFEH can be achieved in a near field by magnetic resonance coupling [52] or inductive coupling [53]. Energy transfer during magnetic resonance coupling is achieved by the activity of two coils that resonate via magnetic coupling at the same frequency. This is in contrast to inductive coupling where energy transfer is achieved through the evanescent wave coupling between two systems that resonate at the same frequency. In such systems, the key features are the incident power density and the conversion efficiency. However, the coupling coefficient *K* is essential in calculating the PCE [53]. The distance of power transfer is restricted, due to coupling coefficient dependency, to the distance of the coils. In addition, the configuration requires suitable standardization and arrangement between the coils. Equation (Equation 4) is used to calculate the coupling coefficient *K* [54].
(6)$$K=\frac{M}{\sqrt{L_{1}L_{2}}}$$
where *M* is the mutual inductance, and $L_{1}$ and $L_{2}$ refer to the self-inductance of coil 1 and coil 2, respectively.

Due to varying rate of attenuation, the near-field inductively coupled power propagates for a shorter distance than the far field power. The attenuation rate of the far-field propagation is 20 dB/decade, which is lower in strength than the near-field energy [55]. Multiple applications such as wearable electronics and mobile phones adopt inductive coupling wireless charging, whereas magnetic resonance coupling is utilized to charge consumer electronics and commercial appliances.

Far-field energy transfer is achieved when the distance of RF power transfer is more than λ/2π and enables a wide area of coverage [49]. Nevertheless, low input power and the distance between the transmitter and receiver would limit the conversion efficiency [56]. The main challenge with this method is boosting the DC power level at the rectenna’s output without boosting the transmit power, as well as energizing devices that are tens to hundreds of meters away from the transmitter [57]. As such, numerous technical efforts have been devoted to design an efficient rectenna. Table 2 compares near-field and far-field energy transmission methods.

### 2.1. Antenna Design and Characteristics for RF Energy Harvesting

The receiving antenna plays a pivotal role in the RFEH system. The antenna is responsible for capturing the received RF signal from the extended sources and significantly impacts the conversion efficiency. The optimization of an antenna to resonate can be at single or multiple frequency bands. A high gain receiving antenna design is required to meet the requirement of high-power input for standard applications such as recharging the mobile phone battery remotely, waking up sensors in a sleep mood, and energizing the wireless sensor network [59]. The antennas need to supply sufficient power for the applications from limited ambient signals. The desirable characteristics in the antenna design for RFEH are high gain, wide bandwidth, miniaturization while upholding the efficiency, and circular polarization for minimum mismatch loss. In biomedical applications, several other aspects need to be considered in designing wearable and implantable antennas, such as flexibility, stretchability, lightweight, and safety. Therefore, this subsection discusses the design characterization and specification of wearable and implantable antennas that are used in the RFEH system.

#### 2.1.1. Wearable Antenna Design for RF Energy Harvesting

The integration of an antenna in wearable medical devices must be conformal and nonobstructive. The mechanical deformations of the wearable antennas affect the bandwidth and center frequency, especially when the antenna operates in a narrow band, where even a small change could detune the frequency of optimization. The wearable antenna must be isolated from the human body loading effect and absorptive losses by the ground plane. Different types of dielectric and conductive materials are used to construct wearable antennas. The selection of these materials is driven by their ability to offer a sufficient number of mechanical deformations (twisting, bending, and wrapping) with little influence from various environmental factors (rain, snow, ice, etc.), as well as adequate protection from EM radiation. Recently, several types of fabric and nonfabric materials have been used to fabricate wearable antennas; however, these fabric materials need proper characterization before being used for such purpose [60]. Given that textiles are widely used and available, several wearable antennas for RFEH are fabricated based on textile materials such as felt fabric, polyester cotton, and Cordura [61,62,63]. These textile antennas are designed with different topologies such as monopole antenna and patch antenna, operated with a single band, and dual-band. On the other hand, numerous wearable antennas for RFEH used nonfabric flexible polymer-based materials due to their stable dielectric properties. For example, the work reported by [64] proposed a dual-band dipole antenna printed on Kapton, while in [65], a monopole antenna printed on a polyethylene terephthalate (PET) film. A dualband transparent patch antenna fabricated on polydimethylsiloxane (PDMS) substrate was reported by [66].

Furthermore, wearable antennas with potential use in RFEH must be stretchable to accommodate natural skin deformations or large human motions [67]. The detuning effect of the stretchable antennas significantly reduces their resonant frequency when they are stretched [68], thereby restricting their usage in strain sensing [69,70,71]. Additionally, their integration with the commercially available chips is difficult. Besides that, the radiation efficiency of the stretchable antennas degraded significantly due to the lossy human tissues [72,73]. Consequently, stretchable antennas with high performance and insensitive resonance frequencies are required for RFEH and communication [74,75,76,77,78,79]. In this context, Zhang et al. [80] designed a stretchable microstrip antenna with different 3D layouts for RFEH. The proposed antenna was fabricated by patterning a meshed patch attached with an elastomeric substrate (Ecoflex00–30) and a meshed ground with an ultraviolet laser. The 3D microstrip antenna achieved better stretchability, strain-insensitive resonance, and improved peak gain when compared to its 2D equivalent. However, further optimizations in the design of asymmetric 3D structures are required to achieve high on-body performance resistant to various mechanical deformations. Conventional and stretchable metals are suitable radiation elements due to their inherent electromagnetic properties and high radiation efficiency. Thus, the work reported [81] proposed a laser stretchable wideband dipole antenna fabricated using conductive laser-induced graphene (LIG) patterns; the surface was selectively coated with a metal and mounted on an elastomeric substrate. The radiation performance of the stretchable antenna was evaluated depending on its tensile strain using a modified stretcher to reflect its combined mechanical–electromagnetic properties. In addition to stretching, the performance of the stretchable wideband dipole antenna is resistant to deformation modes such as twisting and bending.

The safety of the wearable antenna must be taken into account during the design phase, in addition to ensuring that a wearable antenna can blend in with the body. The specific absorption rate (SAR) is used to evaluate the electromagnetic safety of wearable antennas. The International Commission on Non-Ionizing Radiation Protection and the Federal Communications Commission (FCC)/International Electrotechnical Commission (IEC) limits the maximum general exposure to SAR to 1.6 W/kg (SAR 1 g) or 2 W/kg (SAR 10 g), respectively. SAR is computed by averaging over a sample volume, which is typically 1 or 10 g (ICNIRP), respectively.

#### 2.1.2. Implantable Antenna Design for RF Energy Harvesting

Implantable antennas are an essential element in electromagnetic energy conversion, whether they are utilized for RFEH or data transmission in IMDs [82]. Additional difficulties beyond those of wearable antennas must be considered in the design of the implantable antennas as they will be situated inside the body. Body tissues and fluids which act as dielectric loading to the antenna are the primarily limiting factor, adversely degrading the performance of the implantable antennas. Miniaturization techniques are important in the design of implanted antennas since they can minimize their size as much as possible, increasing the possibility that they will integrate with the body. According to various scenarios, researchers have proposed numerous typologies of miniaturized implantable antennas for RFEH. The work reported by [83] developed a circular dual-band implantable antenna with a radiating slot patch; this antenna has an additional external metallic reflector that was positioned behind the human arm to improve its power transmission link. The size of the radiating metallic patch was 10.8 mm, the same as the substrate and the superstrate; this is to ensure that it would touch the surrounding tissue. A dual-band compact planar inverted F-antenna (PIFA) for RFEH was designed in [84]; the developed PIFA is equipped with a matching layer on the arm to improve the wireless power link. The size of the antenna is 16 × 14 × 1.27 mm${}^{3}$. The dual-band operation and miniaturized size of the antenna were facilitated by the use of the slit/slot loading techniques on the radiator. Another study by [85] described a self-diplexing implantable antenna for RFEH antenna that had two ports and a minimal size of 9.4 mm${}^{3}$. A common ground plane connects the two semicircular patches that make up this antenna. However, the complexity of human tissues provides that the aforementioned antennas have small impedance bandwidths; the antennas used in IMDs should have a broad bandwidth to prevent straying outside of the required band. Furthermore, antenna with wider bandwidths often present more RF power at different frequencies. Thus, the authors in [82] reported an implantable, broad dual-band antenna for RFEH. The design of the antenna involved the introduction of multiple radiating branches and etching of a C-shaped slot; multiple resonance frequencies were also produced to accomplish dual and broad bands. The size of the suggested antenna was 7.9 × 7.7 × 1.27 mm${}^{3}$. The circular polarized (CP) implantable antennas are recommended for various biomedical applications due to less polarization mismatch, minimizing multipath interference, and improvement of bit error rates [86], but when there is a need for compact size, greater performance, and biocompatibility, the design of a CP implantable antenna becomes significantly more challenging. In [87], they reported the use of C-shaped slots and a complementary split-ring resonator (CSRR) for the miniaturization of a wide-beamwidth CP implantable antenna; the size of the antenna was 8.5 × 8.5 × 1.27 mm${}^{3}$. A miniaturized CP implantable antenna was designed in [88] for far-field wireless power transmission (WPT). The antenna has a size of 11 × 11 × 1.27 mm${}^{3}$ and features stub loading and capacitive coupling among the stubs. A CP implantable antenna for WPT was reported by [89]. The antenna has a miniaturize size of 7.5 mm × 7.5 mm × 1.27 mm${}^{3}$, designed by etching four C-shaped open slots on the patch. Shaw et al. [86] proposed a wideband, biocompatible, flexible CP slot antenna using a single-layer skin tissue model; the antenna was designed for use as a receiving (Rx) element; the CP antenna has a reduced size of 12 × 12 mm.

Similar to miniaturization, biocompatibility is another important aspect that should be considered during the design of implantable antennas. Biocompatibility can be defined as the capability of the material to work with proper host response in a particular application. Two methods have been proposed to ensure the biocompatibility of the implantable antennas. The first method is based on covering the implantable antenna with biocompatible materials (PDMS) to accomplish the desirable biocompatible specifications [90,91]. The second method is to use biocompatible substrates such as alumina, zirconia, and Macorin the fabrication of the implantable antenna [90,92].

Although different groups have determined SAR for implantable antennas using different phantoms, input powers, etc., the safe limits established by the IEEE, FCC, and ICNIRP remain unchanged. Implantable antennas have similar SAR values to wearable antennas; however, the depth of the implantable device inside the body is a further degree of freedom for SAR determination [93].

A comparative study along with performative analysis of wearable and implantable antennas for RFEH are presented in Table 3, taking into account antenna characteristics: size, realized gain, operating frequency, SAR, and power conversion efficiency.

### 2.2. Impedance Matching Network

The impedance matching network is employed to prevent power leakage, and it guarantees maximum power transmission between the source of energy and the load. To achieve maximum power transfer, the impedance of the output antenna should be identical to the load impedance. The incident wave at the load will be reflected when an impedance mismatch leads to reduced efficiency. In the RFEH system, the receiver antenna acts as an RF source, while the rectifier represents the load. The impedance matching network concurrently functions as a low-pass filter in suppressing higher-order harmonics generated by the rectifying circuit, which can be re-radiated by the antenna, causing additional loss [103]. A perfect matching network for the RFEH should achieve impedance matching between the source and the load with minimum insertion loss and at any input power, frequency, and load resistance. Matching networks are classified into three configurations, which are L, Pi, and T configurations, as shown in Figure 4. The commonly used configuration is L, with two components that ease the control process and design. Moreover, the quality factor Q is sustained in the L configuration. In comparison to the L network, the T and Pi matching designs are more complex. Additionally, organizing the T and Pi configurations into numerous stages preserves the final matching outcomes but will alter the Q factor. This method is advantageous for improving passive voltage boosting.

### 2.3. Rectifier Circuit Design and Topologies

The rectifier functions by converting the harvested RF signal by the receiving antenna to DC output voltage. The main design bottleneck in the rectifier’s design is to generate DC voltage from low power input. There are two types of rectifiers: diode-based rectifiers and metal–oxide–semiconductor field-effect transistor MOSFET rectifiers. When compared to MOSFET circuits, the diode-based rectifier has a lower forward voltage drop; hence, it is preferred in applications. The ability of low forward voltage diodes to achieve higher PCE makes it an optimal choice. For rectenna applications, Schottky barrier diodes are popularly adopted [104]. The half-wave rectifier is the basic rectifier topology; it is composed of a single diode D1, as depicted in Figure 5a. However, for most applications, a half-wave rectifier is insufficient despite its simplicity of design. Therefore, the full-wave rectifier is preferred. Figure 5b depicts the circuit design of a full-wave rectifier. Alternatively, the bridge rectifier (depicted in Figure 5c) exhibits better reverse breakdown voltages, higher power efficiencies, and fewer output ripples compared to its counterparts [105].

Applications driven by high power need are enabled through a rectifier, known as the voltage multiplier, in boosting of AC to DC conversion. The voltage multiplier encompasses numerous single rectifiers connected in series [106]. The Cockcroft–Walton voltage multiplier is the commonly used topology (depicted in Figure 5d). This rectifier is similar to a full-wave rectifier, which integrates numerous stages to achieve higher voltage gains. A modified version of the Cockcroft–Walton’s configuration is the Dickson multiplier, shown in Figure 5e, in which a shunt capacitor is integrated to minimize parasitic effects. The Dickson multiplier is popularly considered for low-power devices despite the tradeoffs in high power conversion efficiency due to the leakage current by the high threshold voltage within the diodes. Furthermore, the output voltage shows a significant reduction under high resistive load, thereby reducing the current supply. Nevertheless, the MOSFET-based diode has a fast switching speed but suffers from electromagnetic interference and thermal runaway; a high threshold voltage is also needed, restricting RFEH circuits’ performance [107]. The Dickson multiplier is suitable for low-power applications enabled by MOSFET technology; it can be realized in system on chip (SoC) solutions by substituting the diode with a negative channel metal oxide semiconductor (NMOS), as depicted in Figure 5f.

## 3. RF Energy Harvesting Application in Medical Devices

The widespread availability and the diversity of RF energy sources (external and ambient) and the exponential growth in Internet of Things (IoT) and wireless sensor nodes has encouraged recent reported research work to employ RFEH techniques to energize low-power wearable and implantable medical devices. Biomedical devices gained widespread application in healthcare sectors due to the favorable feature of continuous monitoring of vital signals from the human body or apparent spurs from the internal body organs. RF energy is a prominent integral block for wearable and IMDs in sustaining the lifetime of the battery in use for an extended period, relaxing the need to change or recharge the battery frequently. This is enabled through the development and integration of ultra-low-power electronics, which decreases the power consumption of the primary sensor components to less than a milliwatt [108]. The current power consumption range of the wearable and IMDs is (nW to mW), as shown in Table 4. Therefore, RFEH technology is an appropriate solution to power these devices. In this section, we will discuss the implementation of the RF energy harvester in wearable and IMDs.

### 3.1. RF Energy Harvesting for Wearable Medical Devices

Wearable medical devices contribute significantly to improving the current medical system and supporting physical activities due to their ability to offer extensive monitoring of different physiological functions, emergency alerting, and computer-aided rehabilitation. These devices can be used in the healthcare sector and military sector for evaluating the mental and physical health of soldiers in combat [118,119]. This current global pandemic outbreak has significantly impacted the global healthcare system and, as such, has necessitated the deployment of remote monitoring systems to monitor its symptoms and perform essential tracking [120]. Researchers have over the years introduced several IoT-based devices and wearables for accurate diagnosis and monitoring of patients in the prodromal phase; these devices can monitor the temperature, respiratory rate, and heart rate of the patients and provide real-time data for accurate treatments. Such devices are low in power consumption (ranges from pW to μW) and can be efficiently powered using ambient RF sources. Low-power devices include temperature sensor devices (power consumption ranges from 113 pW–1.4 μW) [97,121,122,123,124], biosensors, voltage, and current sensors (power consumption ranges from 9.3 nW–436 μW) [125,126,127]. Suggestions have been made for ambient RF power harvesting to power these devices instead of batteries [128,129]. RFEH systems provide a controllable and continuous source of energy. However, the implementation of RF harvesters to power wearable devices faces some issues, such as their ability to supply an adequate amount of energy and the deterioration in power transfer efficiency of far-field RFEH systems. Several innovative methods to design RFEH systems for energizing various types of low-power wearable medical devices are reviewed in the following subsections.

#### 3.1.1. Wearable Inkjet-Printed RF Energy Harvester

The environmental friendliness and low-cost fabrication processes of additive manufacturing, such as 3D printing and inkjet printing, have increased its industrial relevance, as these emerging fabrication techniques are designed to significantly reduce the required number of manufacturing steps, such as the elimination of the etching processes, thereby improving the fabrication efficiency. Most of the time, wearable sensor devices are preferable in areas where disposable or single-time use is hygienically sorted, such as in hospital settings. This application type requires the creation of numerous circuit components within the device, using additive manufacturing to reduce the device cost. The improvements recorded recently in fabrication and performance have endeared inkjet printing technology to sensor and RF applications. In this regard, a design of enabling near-field RFEH for wearable sensors with an additional fabrication process was proposed in [130]. The work fabricated circuit prototypes through a combination of conductive traces developed with conductive inkjet printing and masking technologies with lumped circuit components. Furthermore, the S-parameters were used to estimate the input power for the RF to DC conversion circuit; the circuit produced a peak output power of 0.146 W with a H-field harvester and 0.0432 W with an E-field harvester. The harvesters (E- and H-field) were subjected to several operation tests using a microcontroller communication module and an LED in both on-bottle and on-body bent/flex conditions to validate the functionality based on the energy from a two-way talk radio. The outcome of these tests ensures the compatibility of the developed inkjet-printed flexible energy harvesters to be adopted in wearable biosensors. Besides that, the work in [131] presented a flexible and wearable energy-autonomous on-body sensor network with complete RFEH operability using a handheld 464.5 MHz UHF band two-way talk radio, which was created using 3D printing and inkjet processes. The system is equipped with two different types of energy harvesters; the first one was attached to the sensing-capable backscattering RFID tags for the harvesting of the 464.5 MHz signal energy that will power the tags; the second type of energy harvester was attached to the hands of the wearer for the harvesting of the same 464.5 MHz signal to provide both the carrier signal and the DC power. The efficiency of this second energy harvester is higher than that of standard ambient energy harvesters. Being that the rectifier’s DC and second harmonics were used to support two extra functions, energy harvesting designs were used in the system. The estimated DC power of the energy harvester is 17.5 dBm, while the second harmonic output (SHO) is 1.43 dBm when a two-way talk radio is placed at a distance of 9 cm. Using the harvester-powered RF amplifier, the measured SHO power is boosted to 13 dBm.

To achieve particular structures, traditional 3D printing technologies often demand a large amount of time and support material. Therefore, the work by [132] proposed a fabrication approach to design an origami RF harvester system based on origami folding principles for use in high-frequency applications. This approach was devised such that it considerably decreases the amount of time required for the fabrication and removes the necessity for support material. The method entails fabricating a planar structure using 3D printing technology and then using inkjet printing to directly produce conductors on its surface. The inkjet-printed on-package conductive features were manufactured successfully and integrated with RFEH electronics to demonstrate the effectiveness of using origami techniques to build completely 3D RF systems. At the input power of 0 dBm, the RF energy harvester achieved a DC output voltage of 1.2 V. Furthermore, 3D printing can be a good way of solving huge parasitic problemsfor traditional package structures that have a major impact on the performance of the system [133,134]. An embedded-on-package 5G RF energy harvester that operates at 26 GHz was presented by [135]; this system performs within the 3D printed multilayer flexible packaging. The entire system is made using low-cost, quick-to-prototype additive manufacturing processes such as 3D printing and inkjet printing. In this system, the use of the isotopically radiated power EIRP that is permitted for 5G communication (which is 75 dBm) [136] rules out the issue of low power density for RF energy [137]. At a 20 cm gap from the source, an energy output voltage of 0.9 V was harvested with a transmitted equivalent (EIRP) of 59 dBm, whereas a >1 m range is anticipated when applying the full 75 dBm EIRP for 5G communication. Another work by [138] used a fabrication process that relies on a masking process that is inkjet-printed and then etched. The work focused on improving the capability of typical shunt Schottky diode topologies that are deployed for energy harvesting in rectenna systems. This was accomplished by providing both flat and high-power conversion capabilities across the whole frequency range of interest. The system is a dual-tapered transmission-line-based matching network for the improvement of the rectification capability of the embedded Schottky diode. The rectenna is made up of a rectifier and a tiny monopole antenna that is optimized for a liquid crystal polymer (LCP) substrate. At an input power of 0 dBm, the flexible rectenna achieved a maximum PCE of 40%.

#### 3.1.2. Textile-Based Wearable RF Energy Harvester

Wearable devices may benefit considerably from embedded tiny electronics and conductive materials to usher in the smart clothing paradigm [139]. Rectennas made of textiles can be used in body-centric wireless communication systems that do not require batteries. Studies have reported the fabrication of many antennas made of textile materials for wireless communication [140,141]. To create a wearable device for power harvesting, some works have paired a textile antenna with a rectifier circuit manufactured on a printed circuit board (PCB) [61]. Several rectennas have been designed for single- or dual-band operation. For example, the work reported by [142] described a single-band textile rectenna that operates in the 4.65 GHz frequency band made with jeans cotton as the substrate, while the radiating element is copper tap. The textile rectenna achieved a DC output voltage of 400 mV with a maximum PCE of 55% at an input power of −5 dBm. A dual-band operating textile rectenna was developed by [143] in which the antenna is a sub-1 GHz (785–875 MHz) broad-beam rectenna with a 2.4 GHz off-body antenna. The PCE of this rectenna was 63.9% and the DC output was 650 mV from a power density of less than 0.8 μW/cm${}^{2}$. Next, in [144], a design of a complete textile-based rectenna system was presented. This rectenna is laden with a novel power management platform enabled independently through RF power harvesting. This rectenna operates at a frequency band of 900 MHz and 1800 MHz, and the Wi-Fi band at 2.4 GHz. The RF properties of the textile rectenna are verified through nonlinear techniques, where textile materials and antenna layout are numerically characterized through electromagnetic simulations. The system is entirely autonomous and is operational even without the use of a battery at a low RF power level of −15 dBm.

To increase the output power of a rectenna, numerous research efforts have been devoted to designing a textile rectenna array. The authors in [145] introduced textile-based rectenna arrays for the provision of power in wearable electronic devices. The system is operated at 2.45 GHz and consist of 2 × 2 and 2× 3 rectenna components. Each of the elements consist of a rectifier and a patch antenna created from fabrics using conductive thread embroidery. At −10 dBm input power level, this rectenna achieved a PCE of 35%, whereas the average DC power of the 2 × 3 rectenna was roughly 80 μW at 60 cm from the source. However, the patch antennas and rectifying circuit were on the same layer, resulting in a considerable element-to-element distance. Thus, Chi et al [62] presented a 2 × 2 wearable textile-based rectenna array with a stack architecture for each rectenna to minimize the dimensions of the elements in the array. The rectenna element comprises a linearly polarized patch antenna and a single-stage full-wave Greinacher rectifier which are fabricated using Cordura textile material. The rectenna array was positioned on the human body at a distance of 150 cm away from the indoor access point (Wi-Fi). The maximum DC output voltage of the rectenna array was 1.05 V at an input power of 20 dBm. Juan et al. [146] developed a 2 × 2 textile rectenna array elements for RFEH based on the combination of the electromagnetically coupled microstrip patch antenna and a simple and precise construction method to improve the performance of multilayer microstrip textile patch antennas. Therefore, the patches and rectifier circuit pads were laser-cut, and the various layers were joined together with double-sided, thermally activated adhesive sheets. The patch antennas and the rectifiers were fabricated on pure copper polyester taffeta fabric. The on-body performance of the 2 × 2 textile rectenna array was verified through perfect alignment with the Tx antenna, as depicted in Figure 6. The on-body measurement of the proposed system obtained a maximum output power of 1.287 mW with a maximum PCE of 40% at an input power of 12 dBm. Estrada et al [147] proposed 16- and 81-element broadband rectenna arrays based on a strongly coupled bowtie antenna screenprinted on a cotton T-shirt for harvesting power densities of 4–130 μW/cm${}^{2}$ between 2 and 5 GHz. The broadband rectenna achieved a PCE of 32% at an incident power density of 100 μW/cm${}^{2}$.

#### 3.1.3. Stretchable and Flexible RF Energy Harvester

Due to the frequency detuning caused by the mechanical deformations, such as stretching or bending, the performance of the wearable rectennas was significantly reduced. For the target frequency bands of ambient wireless energies, the antenna size within the rectenna is restricted to a specific range even though the miniaturized, thin-film rectifying circuit design and the matching network can offer a strong mechanical property. Furthermore, for low ambient RF energies (less than 1 mW), the conversion efficiency of the existing rectennas from any single frequency is dramatically lowered [148,149,150], partly due to the limiting diode’s properties and the high loss of the antenna circuit and the impedance matching circuit [150]. Designing and demonstrating high-performance stretchable wideband rectennas and wideband antenna to integrate received RF energy over their bandwidth upon deformations is highly important. In this context, a design of stretchable wideband rectenna to resist mechanical deformations, operate robustly, and incorporate supplied RF power throughout their wideband operation was presented by [81]. The design proposes a hybrid of electromagnetic and mechanical approaches with a fabrication based on the laser to implement a highly efficient stretchable wideband rectenna with compatible mechanical properties. The integration of the rectenna is enabled through RF power with different LIG sensors producing a novel category of stretchable all-LIG devices for remote sensors and wearable devices. The rectenna reduces power consumption due to inherent highly conductive radiations. The rectenna comprises a wideband antenna integrated with a full-wave Greincher rectifier and two matching circuits to enhance the efficiency and sensitivity at the frequency band of 1.75 to 2.45 GHz. A microwave oven is employed as a radiation source in this experiment. The radiation from the microwave oven, which acts as the input power, is programmed to decrease as the distance between the transmitter and receiver increases. Considering the antenna loss and the matching network, this antenna exhibits PCE of 10% at the maximum input power of 0.001 mW. The stretchable wideband rectenna can harvest RF power from a mobile base station against multiple deformation modes such as twisting and stretching. The work reported by [80] designed a standalone stretchable RF energy harvester. The system comprises asymmetric 3D microstrip antenna incorporated with a matching network and flexible rectifier circuit forming a stretchable rectenna system to harvest the RF energy. The stretchable rectenna achieved a longer energy transfer distance and a double charging rate from the captured RF energy due to the optimized peak gain of the asymmetric 3D microstrip antenna. The standalone RF energy harvester consists of a stretchable rectenna integrated with stretchable sensing and energy storage module. To test the system in a practical application, the microstrip antenna was incorporated with the flexible supercapacitor and a stretchable strain sensor, both made of LIG foam. The stretchable rectenna efficiently charged the flexible supercapacitor from 0 to 0.5 V within 200 s. The stored energy in the supercapacitor was able to produce a DC output voltage of 1.8 V to energize the strain sensor.

Numerous rectenna systems have been described with features such as flexibility, lightweight, compactness, and wideband operation. Palazzi et al. [151] presented a new, compact, ultra-lightweight multiband RF energy harvester built on a paper material that covers all the recently released LTE bands (0.79–0.96 GHz; 1.71–2.17 GHz; and 2.5–2.69 GHz). The use of nested annular slots topology can offer great compactness and easy combination of rectifier and antenna. In the studied bands system, the suggested rectifier recorded PCE of 5–16% at 20 dBm input power, which improves to 11–30% at 15 dBm. Next, the study published by [148] constructed an atomically thin and flexible rectenna that was fabricated from a MoS2 semiconducting–metallic-phase heterojunction with 10 GHz cutoff frequency, which is a significant speed improvement of about one order of magnitude. The same piece of Kapton film was also used to create the MoS2 rectifier and flexible receiver antenna. RF power was wirelessly harvested in the Wi-Fi channel (5.9 GHz) using the flexible MoS2 rectenna which produced up to 250 mV rectified output voltage at a distance of about 2.5 cm from the transmitter antenna. At 0.7 dBm input power, this rectenna achieved a maximum PCE of 40.1% at 2.4 GHz.

A flexible RFEH system’s low efficiency can be solved by introducing multipath in the environment while quasi-omnidirectional RF reception can be achieved using antenna diversity; this increases the chances of receiving energy in a realistic environment. The study by [152] proposed a 3D flexible antenna diversity that can be fitted into rectangle packaging with the aim of exploiting the benefits of the packaging form andambient RFEH operation in an indoor environment. To ensure low cost and flexibility, Rogers 4003 substrate was used to design the angle and polarization diversity antenna; this resolved the issue of poor substrate performance and preserved the antenna’s compact size, strong isolation, and high radiation efficiency. Various scenarios were performed for the measurement of rectenna in a realistic environment. Scenario S1 tested the available power in the environment at each point and direction by employing a typical antenna patch. Scenario S2 evaluated the absorbed power at each point and in each direction, utilizing the typical antenna patch and rectifying circuit A (Rectenna 1A). Scenario S3 employed four patch antennas oriented in opposite and perpendicular orientations to exploit the geometrical advantages of the packing box connected with Rectifier B (Rectenna 1B). This scenario was designed to assess the performance of a varied antenna system operating in an interior environment. Scenarios S4 and S5 used the flexible antenna diversity with a separated rectifier (Rectenna 2) and an integrated rectifier (Rectenna 3), respectively. These scenarios compared the performance of the flexible rectenna diversity and conventional patch diversity. The proposed 3D flexible structure achieved a median harvested power of 124 nW, which corresponds to a nominal PCE of 6.1% at 0.1 mW/m${}^{2}$ median input power. Most of the work described above focuses on creating a single rectenna that may not be powerful enough for certain applications. Therefore, the work by [153] presented a cylindrical dual-band flexible rectenna array that works in the LTE band (1.8 and 2.4 GHz) which can harvest RF energy from a range of sources within the azimuth plane. The components of each rectenna subsystem include a monopole antenna (dual-ring-shaped dual-band) and a rectifying circuit (dual-band) fabricated on a polyimide substrate. At 12 dBm input RF power, the designed single rectenna unit can achieve up to 40% power conversion efficiency.

#### 3.1.4. Discussion

The previous section explored various ways of designing RFEH systems for powering a range of low-power wearable medical devices. RFEH technology is an appropriate solution for powering wearable medical devices in order to maintain power autonomy and ensure maintenance-free batteries. There are some issues associated with the design of a complete wearable rectenna, such as the reliability of the connectivity between rigid and flexible system components, optimizing the power conversion performance to enhance transfer range, maintaining efficiency and pattern of radiation on the human body, and the performance fluctuation caused by the bending effect, particularly for the antenna. Therefore, advancements in the design of lightweight, compact, and flexible rectennas capable of harvesting the maximum amount of RF energy from different RF sources randomly distributed in space are still required. One of the possible solutions to achieve this is by designing a flexible and lightweight rectenna array arranged in different orthogonal orientations. On the other hand, future advancement in the design of stretchable rectennas can be achieved by integrating or using the metasurface ground [154,155] or stretchable patch antennas [70] with wideband designs [81,156,157] to improve the on-body RFEH performance and further decrease the absorption in the lossy tissues. There are two approaches to designing highly efficient stretchable wideband rectennas. First, by using stretchable wideband antennas with circular polarization [158] or dual-polarization [159] to incorporate the randomly polarized EM field in a realistic environment; second, by using the spin diode rectifiers to improve the PCE [160].

The performance of the wearable RFEH system is affected by several technological challenges, such as the decrease in available energy density as propagation distance increases; impedance mismatch that is caused as a result of differences in the input resistance and reactance between the antenna and rectifier circuit; decreasing the rectenna size while improving the PCE; the dependency of the PCE on the enhancement of the RFEH circuit sensitivity. To overcome these challenges and improve the performance of the wearable RFEH system, several considerations need to be accounted for, such as the following:The efficiency of the system determined by the suitable range and operating frequency;The design and integration of an appropriate rectifier circuit by evaluating the required output power and input sensitivity;The integration of a compact matching network for maximum power transfer independent of input power, frequency, or load change;The received signal strength is dependent on the design of an antenna with high gain and wide bandwidth;The design of a flexible and lightweight rectenna, capable of harvesting RF power from numerous RF sources, distributed randomly in the space;The antenna used in the wearable rectenna design should withstand various mechanical deformations such as stretching, bending, rolling, and twisting.

Table 5 compares the performance of different RFEH systems for wearable medical devices. These systems are capable of powering various wearable medical devices, such as wearable health monitoring sensors and smartwatches. From this comparison, we can understand that the distance from the source and operating frequency significantly impact the harvested energy and PCE.

### 3.2. RF Energy Harvesting for Implantable Medical Devices

The highlight of IMDs is in the continuous monitoring of human body biological signals to enhance healthcare quality. Due to the favorable inherent low power consumption of IMDs, the goal of recent research works is to extend the battery lifetime of the devices in use. Many of these devices are enabled through a battery that needs frequent replacement, and in the case of invasive devices, surgical intervention is necessary which affects the comfort of patients. Supplying the IMDs with stable and permanent renewable energy sources remains an ongoing challenge. Ambient RFEH is considered as an appropriate solution in addressing this bottleneck. RFEH for IMDs can be achieved in the near-field or far-field scavenging. Some works have also reported midfield energy harvesting [85,165,166]. Therefore, the subsequent part of this paper discusses the RF energy harvesting for IMDs in the near field, midfield, and far field.

#### 3.2.1. Near Field

In the conventional near-field inductive and resonant coupling [167,168,169], an optimized design of a transmitter coil is important to increase the wireless power transfer (WPT) efficiency. Power leakage is an additional impediment restraining the widespread use of WPT. The power leakage potentially causes failure to other medical devices [170], posing serious health risks. The conventional inductive coupling method can contribute to the focusing of magnetic field in a specific focal region within human tissue. For instance, the low-frequency inductive coupling has been used for wireless telemetry of IMDs since lower frequencies have better penetration in the tissues [171]. Despite the wide use of near-field energy harvesting by the reported research works, there are some bottlenecks in adopting this system. The system requires consistent calibration, accuracy, and alignment between the transmitter and receiver coils, where the distance between IMDs and the power supply must be only few centimeters. However, studies in [172,173] adopted the radiative near-field (RNF) region (Fresnel zone) of the transmitter Tx, which is less sensitive to Tx–Rx misalignments. The IMDs that use the RNF WPT technique are powered at the operating frequencies of 1.9 GHz and 2.4 GHz. Moreover, the devices are incorporated with an antenna, rather than spiral coils, to capture the RF energy. The work reported in [173] used the RNF technique to eliminate the alignment issue with power of the sensor devices enabled wirelessly from the near-field radiating patch antennas. Shah et al. [174] designed a miniaturized wireless IMD that uses the RNF WPT technology for wireless monitoring elevated intracranial pressure (WICP). The prototype of this device comprises an Rx antenna, a styrofoam spacer, dummy PEC elements realized as a dual layer PCB, and an alumina container. The power transfer efficiency (PTE) of the IMD was measured in terms of distance variations. The maximum achieved PTE was −25.9 dB at 20 mm (0.1267λ) Tx–Rx separation. The setup to demonstrate the performance of WPT comprises an external RF transmitter coil, a receiver antenna immersed in saline solution, a rectifier circuit, and a green LED employed as a DC load. The work reported by [175] proposed a metasurface-based technique to improve the efficiency of the RNF WPT system. The technique is based on using the high refractive index of the metasurface to improve the PTE. The metasurface, which includes a transmitting patch and a receiving implantable loop antenna (see Figure 7a–c), is positioned above the surface of the skin layer in the WPT link. The Rx element is located in a skin-mimicking gel, minced pork, and a pork slab incorporated with a metasurface, as described in Figure 7d–f. The system exhibited a PTE of 1.26% for the skin-mimicking gel. To extend the operating distance in the near-field WPT, the work reported by [176] proposed a synchronized biventricular (BiV) pacing in a leadless fashion by implementing miniaturized and inductively powered pacemakers. An integrated circuit design was employed to reduce the power consumption of these pacemakers, which significantly increased the maximum operating distance to 8.5 cm and 11 cm from 1 W Tx power at frequency bands of 40.68 MHz and 13.56 MHz, respectively. Junho et al. [177] used an inductive coupling method for the WPT system to recharge an implantable ECG monitoring device continuously for 23.6 h. The Tx and Rx coils were fabricated to charge the battery wirelessly, as depicted in Figure 8a. The mainboard, a battery, and the Rx coil were positioned together similarly to the ECG monitoring system, with the coil spacing set at 4 mm, taking into account the animal testing situations as illustrated in Figure 8b. The proposed ECG monitoring device was implanted into a rat to measure and transmit the ECG, as shown in Figure 8c,d. The WPT system achieved a charging voltage of 4.2 V with a PTE of 10% at an input power of 1.8 W. However, the WPT system suffers from electromagnetic interference. The biopotential is small and, thus, it is exposed to noise, such as 60 Hz, from the power line. As a result, the ECG was not measured accurately while utilizing the WPT.

Magnetic resonant coupling is another type of near-field WPT technology that is often used to power IMDs. This method offers high PTE and has been used to power ventricular assist devices; the method used implantable wire coils (length = 9.5 cm, diameter = 22 mm) for energy delivery [168,171]. However, the frequency employed in this technology is quite low, hence limiting the channel capacity and the potential data rate that is required for numerous applications, such as retinal prostheses and recording of multichannel neural networks [178,179].

On the other hand, the study by [180] experimented with a wireless powering method based on near-field capacitive coupling (NCC) for efficient energy transmission to IMDs. The authors identified that the sub-GHz frequency range is the appropriate operating frequency of the NCC technique for subcutaneous power transfer by modeling the power link. The implementation of a conformal and flexible power receiver, as well as compliance with the IEEE C95.1 standard for safe absorption, are desirable features of this method. The NCC link was designed and tested on a non-human primate (NHP) cadaver, and the results show its ability to safely supply power (up to 0.1 W) to IMDs at a maximum efficiency of >50%.

#### 3.2.2. Midfield

The midfield WPT method is an integration of near-field inductive and far-field radiative methods at a frequency range of sub-GHz–GHz. This method was presented to eliminate the limitation of the traditional WPT methods [178,181,182,183,184,185,186]. A conventional midfield method is created based on an appropriate operating frequency and proper current distribution surface on the transmitter source. This method is focused on exploiting propagating fields in the electromagnetic midfield, where the wavelength is identical to the transfer depth. Unlike the near-field scenario, the 3D field pattern in this approach is determined by interference which allows improved energy transfer by manipulating the power flow lines [165,187,188]. In order to obtain high PTE in a deep implantation position, an appropriate operating frequency is selected depending on the depth of IMDs and tissue layer properties [182]. Multiple midfield WPT systems have been presented to power numerous IMDs. Andrew et al. [186] proposed a midfield WPT system that operated at almost any area in the body. A midfield source with numerous excitation ports was developed in this system. The system was evaluated by two configurations that mimic power transmission to devices in the cortex area of the brain in pigs and the left ventricle of the heart. The received power at the implant coil for the heart and brain configurations were 198 μW and 200 μW, respectively, with the Tx source power of 500 mW located 40 mm away. However, this system requires an external controllable circuit to achieve proper distribution of the current. The study by [189] reported the use of a multiband conformal antenna for midfield WPT in IMDs. The antenna was tuned using a T-shaped ground slot before wrapping it with a 3D printed capsule prototype to test run its suitability in various biomedical devices. A shorting pin scheme was suggested to reduce the complexity of the system and obtain a circular current path with two excitation ports. The system was tested in minced pork muscle and obtained an output power of 2.9 mW with a 1 W Tx source power located 55 mm away. Nguyen et al. [183] proposed a compact transmitting design in the midfield band that can concentrate the magnetic field into human tissue. To create a focusing field with one excitation port, an aperture coupled excitation approach was investigated in this work. The system was evaluated in porcine muscle and achieved an output power of 5.6 mW with a 1 W Tx source power located 55 mm away. Notwithstanding, the transmission coefficient of this system is reduced by about 7 dB when rotating Rx, which means the system is sensitive to Rx misalignment. To overcome the misalignment sensitivity of the midfield WPT system, the work reported by [166] designed a bipolar spiral Tx construction. At the Tx source, a multiturn design is used to identify an appropriate current distribution surface and to produce a rotating magnetic field inside human tissue. The system was measured in minced pork and achieved a transmission coefficient of −20.48 dB and −22.15 dB at a distance of 45 mm and 60 mm, respectively.

The efficiency of the WPT system is reliant on the RF-to-DC rectifier since efficient rectification and effective RF power contribute significantly to a successful midfield WPT. Hence, a rectifier can achieve wide operating bandwidth and high conversion efficiency to compensate for the frequency shifts due to differences in the electrical potentials of the body tissues [190]. However, previous studies have shown that it is a difficult task to achieve both extended input power range and broadband characteristics in a high-efficiency rectifier design because a rectifying diode exhibits nonlinear characteristics at different input power levels and operating frequencies. Considering these, the work by [191] proposed a novel rectifier that achieves both broadband and wide dynamic input power range; a modified real-frequency (MRF) technology was used in the design of the rectifier circuit to power deep-placed IMDs in the human body. As a numerical technique, the MRF technique achieves optimum performance by generating a bounded input reflection coefficient, and this differentiates it from the traditional methods where the preselection of the network topology is needed to design the matching network. The operating bandwidth of the fabricated rectifier ranged from 0.9–1.7 GHz, whereas the PCE was >50% at the input power level of 0 dBm. The compact size of the circuit, which is 16 × 11 mm${}^{2}$, made it suitable for use with IMDs. The proposed rectifier recorded a measured DC power at load value of around 0.9 mW upon the application of 250 mW to the transmitter source.

#### 3.2.3. Far Field

As the far-field radiative charging is more resistant to changes in antenna location, orientation, and environment, it is more suitable in the use for higher implantable depth. It is adoptable for high-frequencies’ operation, given that a small supporting antenna can be used to reduce the implanted devices’ size. Hence, the proposed solution is implanted deep inside a patient’s body without much discomfort as the user’s mobility is not restricted while the device is charging (inductive charging is not affected by the position or location of the device). Additionally, RF power transfer is significantly safer than inductive charging because of the rigorous power density regulation for the power transmitter.

Reported research works have investigated RFEH in far field; for instance, the feasibility of implementing far-field RF powering enabled by the RFEH technique was investigated by [192] using an access point (powering frequency = 403 MHz, transmit power = 1 W) for wireless power transfer. The proposed system adopted a harvest-then-transmit protocol to enable power transmission to the implanted device by the access point during the downlink phase. Consequently, the power signal is used for battery recharging and back-transmission of the generated information by the implants toward the access point during the uplink stage. The work in [193] proposed an implanted rectenna system comprising a planar inverted-F antenna (PIFA) and rectifying circuit for WPT to far-field biomedical sensors at 2.45 GHz. The wireless power link was enhanced by adding a parasitic patch onto the human body, which increases the received power level. The study also optimized the PCE after estimating the level of the power received by the implantable antenna based on safety constraints. A wireless optoelectronic system (soft and biocompatible) was developed by Park et al. [194]; the system can sustain natural motions and molds towards their environment to allow operation in areas that were not previously accessible. The components of the proposed device consist of an RF antenna, an LED for neural modulation, and an RF–DC rectifier. The size of the antenna is 3 × 3 mm${}^{2}$ with the ability to receive up to 2.34 GHz of an input frequency. The antenna design should be durable against antenna strains since an excessive strain could decrease the efficiency by 12%. The LED could produce up to 100 μW of optical power when energized with 2 W transmitters distanced about 20 cm away; this eliminates the limitations of midfield and inductive coupling powering techniques. Changrong et al. [88] proposed a CP implantable antenna with a miniaturized size and good radiation performance to develop a power link with minimum polarization. The system was tested in minced pork and obtained a DC power of 5.14 μW from a power source located 0.4 m away. However, the CP implantable antenna has low realized gain due to its operating frequency of 900 MHz.

Analyses of the SA, SAR, and increase in temperature were reported by [195]; the study aimed to verify the impulse radio ultra-wideband transmitter device in compliance with international safety standards. The work in [196] focused on the safety aspect of far-field powering using simplified theoretical analysis and FCC safety limits for radiating antennas. The design of a discrete-transmit-and-receive-on-chip antenna for the use of implantable intraocular pressure monitoring devices was presented by [197], whereas the work of [198] proposed a triple-band implanted antenna capable of data telemetry (402 MHz), wireless power transfer (433 MHz), and wake-up control (2.45 GHz). This triple-band antenna was used in the rectenna for 433 MHz wireless powering transmission, and achieved conversion efficiency of about 86% at 11 dBm input power with a 5 KΩ load. A wireless link operated at 2.45 GHz to power on miniaturized biomedical devices was presented by [199]. A chip antenna was adopted for the transmission of RF power. In this link, the authors used a resonant network that consists of a chip capacitor and compact inductor for energy harvesting.

#### 3.2.4. Discussion

In the former section of this paper, the IMDs whose power needs meet with RFEH technology were reviewed. Table 6 summarizes the characteristics of the recently reported devices. The transmitting method, test model, size, frequency, depth, and transfer distance are presented, along with the harvested power and PTE. In this subsection, we will compare the wireless power transfer methods and discuss the safety regulations and packaging materials for IMDs.

**Wireless Power Transfer Methods**: The near-field inductive coupling WPT method transfers a large amount of power to IMDs with a depth of approximately 10–30 mm. However, the performance of the link in this method deteriorates significantly due to high-frequency losses of the human tissues [204]. Furthermore, the distance between the Tx power source and the Rx implant in this method cannot surpass a few centimeters, and proper alignment is needed between the internal coils and external coils, which is difficult to implement in some cases [205]. On the other hand, the near-field magnetic resonant coupling method provides high PTE. However, the frequency employed in this technology is quite low, hence limiting the channel capacity and the potential data rate that are required for numerous applications. Additionally, this method requires precise alignment and an extra component to be stuck in the skin to overcome the short communication range.

Midfield regime power transfer methods have a wider transmission range, but their efficiency is less than that of inductive coupling methods [165]. However, these devices allow greater implant depth, compared to the previous types of devices, at the expense of the device efficiency. They also require careful alignments between the equipment; the transmitting component of the system must also be equipped with an external power source, and this can affect the mobility of the user.

In the far-field WPT method, the misalignment sensitivity is lower, and the power transfer distance between the power source Tx and the Rx implant is extended to a few meters, which is more convenient for patients. Furthermore, this method can be combined with other RFEH technologies to power the IMDs [206]. However, the PTE of this method is low, compared with the near-field method, due to the loss of the human tissues. Moreover, the SAR regulation limits of this method also reduce their power density.

In conclusion, the midfield power transfer method, which permits focusing on the energy hotspot and overcome the shortcomings of the near-field and far-field methods seems to be the best benchmarked option. Far-field methods are mostly considered when the mobility of patients is the priority at the expense of the efficiency and power level of the link.

**Safety Regulations**: Specific conditions must be fulfilled to design a safe and efficient RFEH system for IMDs. The safety regulations of ICNIRP and IEEE involve general exposure only; the potential field produced by IMDs is described as an untested case [207]. Therefore, special regulations are necessary with a precise assessment of SAR, currents, and E-field in the human body to offer sufficient protection for patients [208]. Both electric and magnetic fields pose a risk of exposure in IMDs that operate with different frequency ranges. In near-field applications, SAR can be used to assess the electrical fields in tissue depending on eddy current, which is the main contributor [209]. When it comes to the high-frequency range, a considerable improvement of local absorption in the skin can be evaluated with standing wave effects [210]. Moreover, inaccurate tissue modeling may increase the difficulties of performing an accurate risk assessment. The complexity of the human body demands various structural and material in vitro model designs. For tissue, which is considered nonmagnetic [211], the conductivity and permittivity properties [212] should be emphasized. Different properties influence tissue absorption and electric field distribution, ultimately affecting the SAR [213,214].

**Packaging Materials**: The packaging material for IMDs is an important factor to assure biosafety. The recommended metals for IMDs could have negative consequences once implanted. To avoid these undesirable consequences, the entire system must be encapsulated in biocompatible material [215,216]. The encapsulation materials used were consistent with the original design of IMDs, such as NuSil and PDMS [181]. However, different encapsulations might have an impact on PTE. In terms of material considerations, metamaterials might boost the electromagnetic field while maintaining the mass augmentation under control [217].

## 4. Conclusions

A review on enabling wearable and implantable medical devices with RF energy is presented in this work. The principles in the construction of the RF energy system are discussed, along with the antenna design and characterization, matching network, and rectifier topologies. Multiple wearable medical devices enabled through RF energy were also analyzed. Notably, RF energy is a reliable and desirable energy source for wearable medical devices; however, implementing the RF harvester in wearable medical devices faces several challenges. For example, the reliability of connectivity between rigid and flexible system components, optimizing the power conversion performance to enhance transfer range, maintaining efficiency and pattern of radiation on the human body, and the performance fluctuation caused by mechanical deformations, particularly for the antenna. Therefore, advancements in the design of lightweight, compact, and flexible rectennas capable of harvesting the maximum amount of RF energy from different RF sources randomly distributed in space are still required. The RF power transfer of implantable medical devices in the near field, midfield, and far field was also presented, with an analysis of the recently published research works that addressed the targeted fields. Midfield power transfer methods for implantable medical devices, which permit focusing on the energy hotspot, seem to be the best choice since the near-field method suffers from a limited performance range while the far-field method suffers from deterioration of the link efficiency. In fact, most of the RFEH devices are not ready for clinical study since they require time, resources, and compliance with strict safety standards for completion. Future advancements in applying the RF harvester in wearable and implantable medical devices are achievable through extensive cooperation between academia and industry to produce an enhanced system that can provide reliable and constant power supplies.

## Figures and Tables

**Figure 1 sensors-22-06371-f001:**
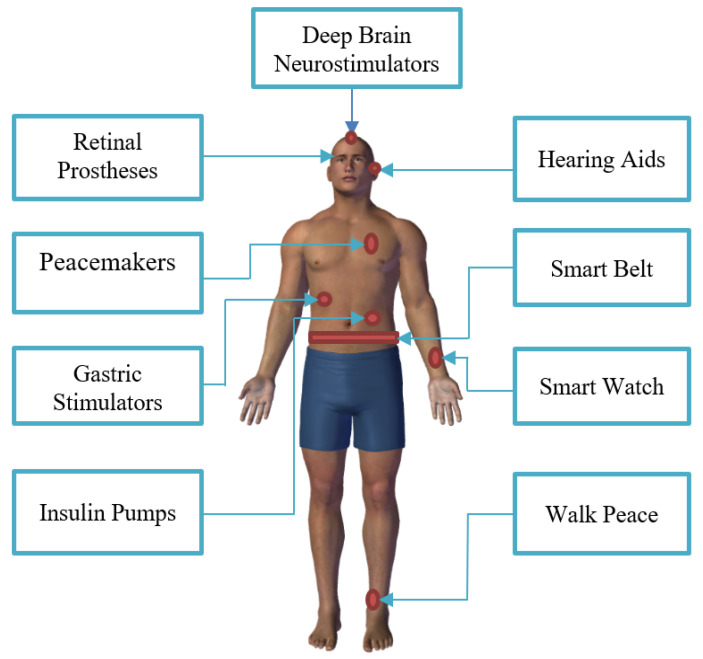
Implantable and wearable medical devices for various body locations.

**Figure 2 sensors-22-06371-f002:**
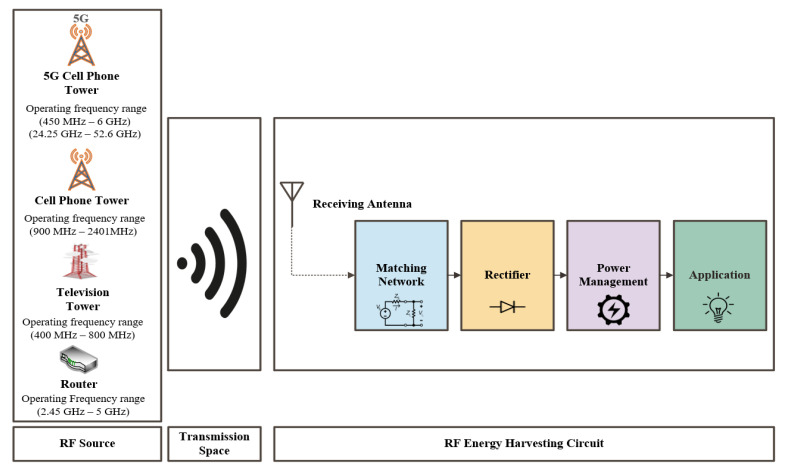
Schematic illustration of the RF energy harvesting system.

**Figure 3 sensors-22-06371-f003:**
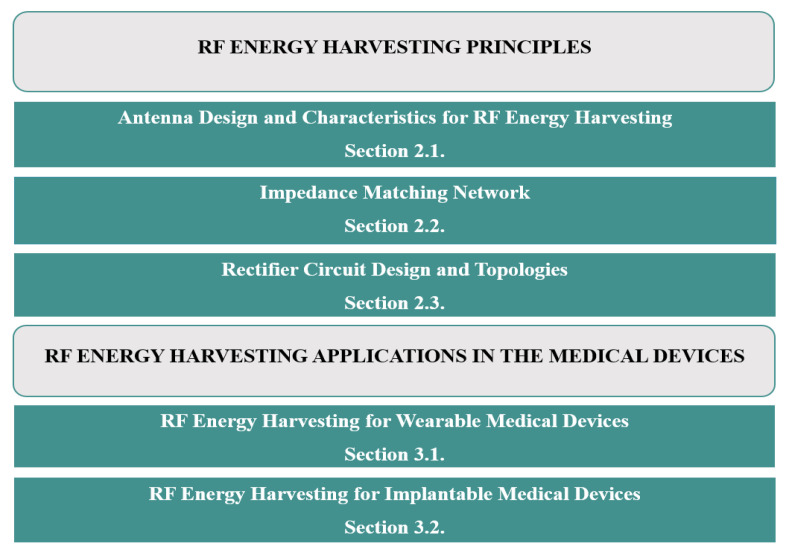
Organization of the paper.

**Figure 4 sensors-22-06371-f004:**
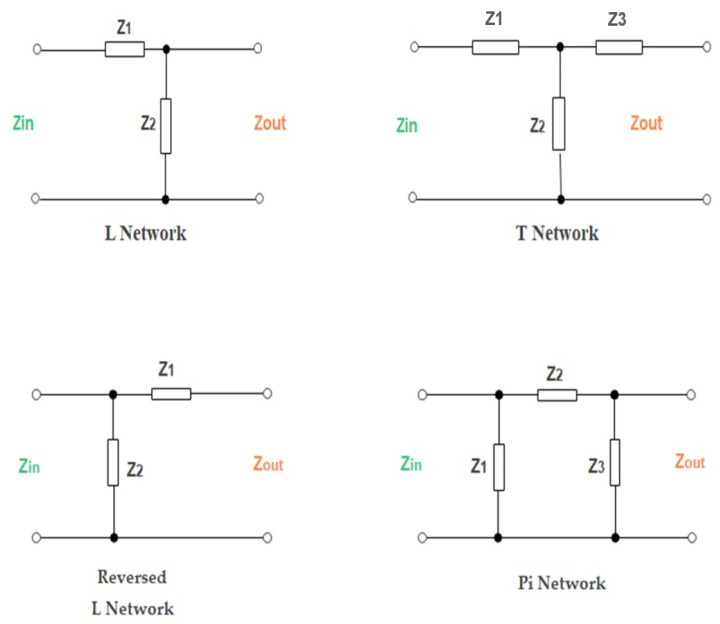
Different forms of common impedance matching networks.

**Figure 5 sensors-22-06371-f005:**
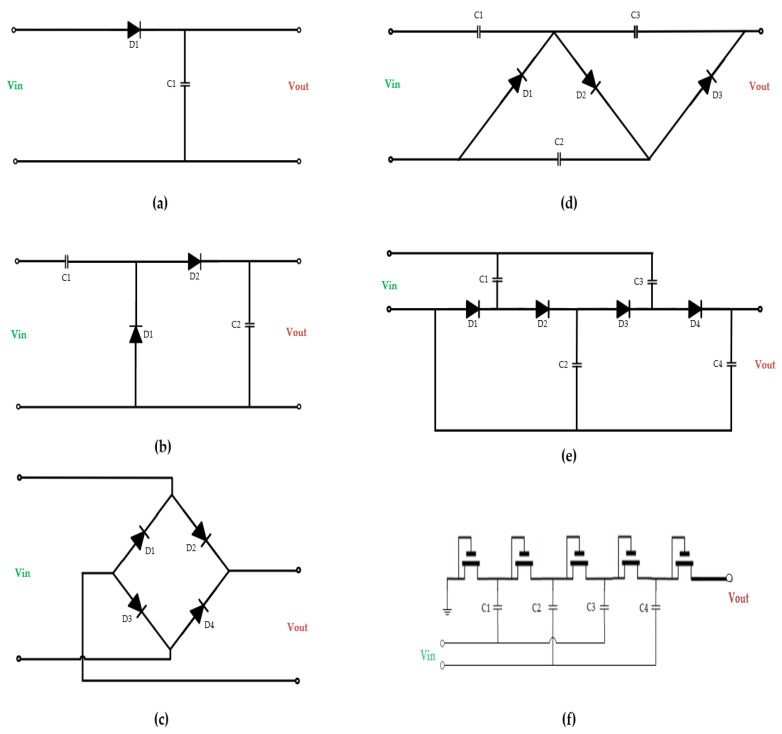
Various rectifier topologies: (**a**) circuit diagram of half-wave rectifier; (**b**) circuit diagram of full-wave rectifier; (**c**) circuit diagram of bridge rectifier; (**d**) circuit diagram of three-stages Cockcroft–Walton voltage multiplier; (**e**) circuit diagram of four-stages Dickson voltage multiplier; (**f**) circuit diagram of four-stages Dickson voltage multiplier utilizing CMOS technology.

**Figure 6 sensors-22-06371-f006:**
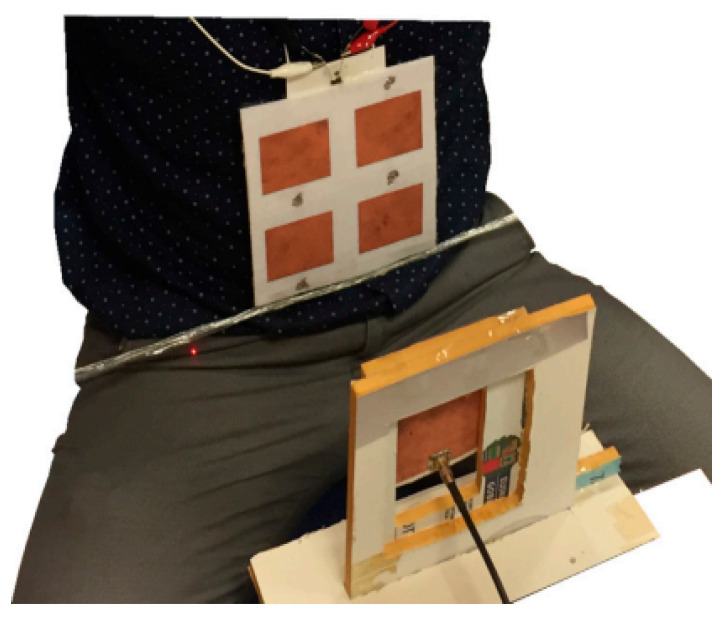
On-body experimental setup of 2 × 2 textile rectenna array [146].

**Figure 7 sensors-22-06371-f007:**
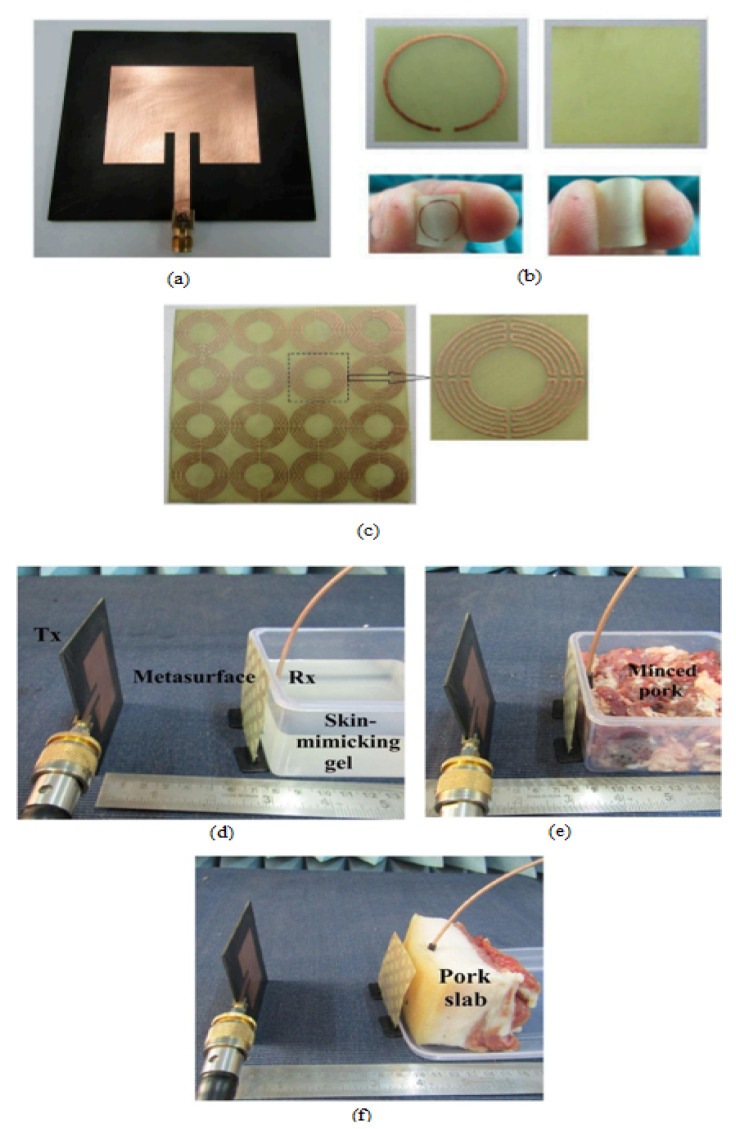
Prototype and measurement setup of implantable WPT system: (**a**) fabricated Tx patch antenna; (**b**) fabricated RX implantable loop antenna; (**c**) metasurface design of 4 × 4 unit cell array; (**d**) testing RX element inside skin-mimicking gel; (**e**) testing RX element inside minced pork; (**f**) testing RX element inside pork slab [175].

**Figure 8 sensors-22-06371-f008:**
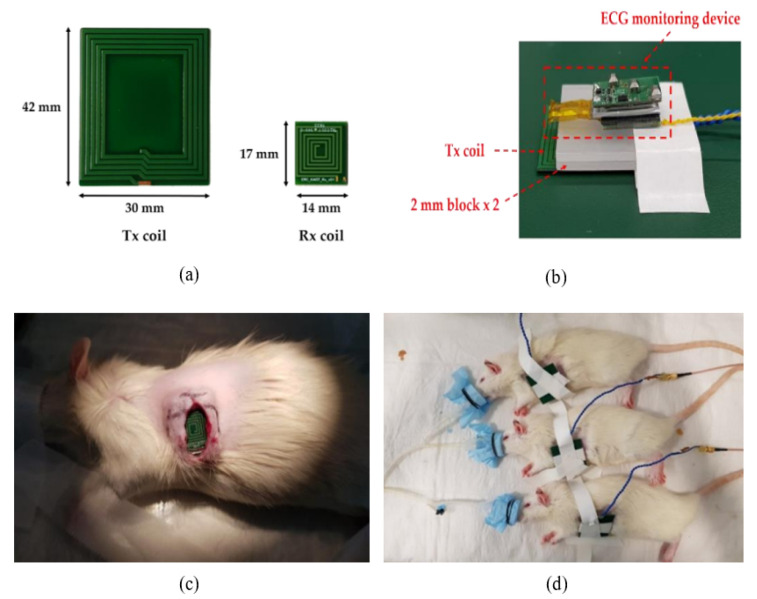
WPT system for implantable ECG monitoring device: (**a**) fabricated Tx and Rx coils; (**b**) mainboard, battery, and Rx coil positioned similar to the ECG device; (**c**) implantation of the ECG device in the rat; (**d**) WPT process in anesthetized rats [177].

**Table 2 sensors-22-06371-t002:** Power transmission characteristics [58].

Features	Resonant Coupling	Inductive Coupling	Far-Field Transfer
Field	Resonance	Magnetic method	Electromagnetic
Scheme	Resonator	Coil	Antenna
Efficiency	High	High	Low to high
Distance	Medium	Short	Short to long
Frequency	KHz to MHz	KHz to MHz	GHz
Power	High	High	Low to high
Typical load	Fixed impedance	Varying impedance	Fixed impedance
Regulation	Under discussion	Under discussion	Radio wave
Pros	Medium efficiency in a short distance	High efficiency	Long distance
Cons	Difficulties in preserving high Q	Very short distance	Low efficiency and safety issues

**Table 3 sensors-22-06371-t003:** Comparison of antenna performance for wearable and implantable medical devices.

Reference	Antenna Type	Size	Substrate	Gain (dBi)	Frequency (GHz)	SAR (1 g Average) W/Kg	PCE (%)
[83]	Implantable dual-band miniaturized circular antenna	10.8 mm	Rogers RO 3210	−23.2	0.42–0.91	0.36	58
[94]	Implantable slot antenna array	30 × 30 mm	Rogers 3010	−26	0.915	175	50
[95]	Silicon Carbide implantable antenna	4.5 × 4.5 mm	Semi insulating (4H-SiC)	—-	10	——	47.4
[96]	Quad-Band Implantable Antenna	8.43 mm${}^{3}$	RO3010	–34, –29.6, –28.2, –22.4	0.403, 0.915, 0.147, 2.4	0.87	0.67
[97]	Broadband Implantable Antenna	91.44 mm${}^{3}$	Rogers 6010	–32, –34	0.72–1.504	921	——
[98]	Compacted Conformal Implantable Antenna	48.98 mm${}^{3}$	Rogers ULTRALAM	−30.8, −19.7, −18.7	0.402, 0.915, 1.2	293.7	——
[99]	Broadband Substrate-Independent Textile Wearable Antenna	0.312 × 0.312 $\lambda^{2}$	Felt and Polycotton	2.2	0.9	1.52	40
[100]	A circular microstrip patch wearable antenna	42.92 × 42.92 mm	Duroid 6010LM	——	2.45	—	25.5
[101]	Wearable Bandenna	35 mm (outer radius)	silicone	5	2.45	——	——
[102]	Folded Dipole Wearable Antenna	0.212 × 0.212 $\lambda^{2}$	Kapton	−0.3	0.94	——	78.5

**Table 4 sensors-22-06371-t004:** Average power requirements and specifications of various biomedical devices.

Reference	Biomedical Device	Size	Power Consumption	Application
[109]	Pulse oximeter	38.69 cm${}^{3}$	294 mW	Measures the oxygen saturation level
[109]	Hearing aid	2.45 cm${}^{3}$	1.82 mW	Amplifies the sound for the patient with hearing loss
[110]	Cochlear implant	9.58 × 9.23 mm	100–2000 μW	Stimulates the cochlear nerve electrically
[109]	Pacemaker IC	4.9 cm${}^{3}$	0.28 mW	Monitors heart rate
[111]	Drug pump for ophthalmic use	9.9 × 7.7 × 1.8 mm	400 μW	Controls drug delivery
[112]	Neural activity monitoring recorder	3 × 3.5 mm	1–10 mW	Records brain activities
[109]	Combo insulin pump	97.61 cm${}^{3}$	15 mW	Delivers insulin using an insulin pump while also monitoring blood glucose levels and giving bolus instructions using a blood glucose meter
[113]	Wireless intraocular pressure monitor	0.5 × 1.5 × 2 mm${}^{3}$	3.65 nW	Frequent measurement of intraocular pressure
[114]	Health monitoring sensor on wristband	——	0.83 mW	Monitors chronic respiratory disease
[109]	Wireless EKG system	40.13 cm${}^{3}$	60 mW	Monitors and records vital signs and cardiac information
[115]	Electrocardiogram amplifier	——	2.76 μW	Transforms the weak electrical signals from the heart into signals that can be transmitted to a monitoring system.
[114]	Electronic-nose sensor system	——	250 μW	Connected to a pattern-recognition system that responds to odors passing over it
[114]	Spirometer	6 × 6 mm	0.01 mW	Measures FEV1, PEF, and FVC
[116]	Cardiac activity sensing	6 mm	0.3 μW	Monitors vital signs
[117]	Retinal prostheses	Diameter = 3mm Pixel size = 25 μM	250 mV	Stimulates the retina
[114]	ECG chest patch	84 × 39 × 8.3 mm	0.96 mW	Measures electrocardiogram ECG, skin impedance, photoplethysmography (PPG), motion, and acoustic signals

**Table 5 sensors-22-06371-t005:** Comparison of wearable RF energy harvester devices.

Reference	Wearable Device	Distance From the Source (m)	Frequency (GHz)	Input Power (dBm)	Harvested Voltage (V)	Max PCE (%)
[62]	Wearable rectenna array	1.5	2.45	−40-0	1.05	——
[144]	Fully-autonomous integrated RFEH system	1	0.9, 1.8, 2.45	−15	3	——
[161]	Dual-band front-end RF energy harvester	——	0.915, 1.8	−33	1	44
[130]	Wearable RFEH from a two-way talk radio	0.07	0.464	17.185	17.87	82.5
[162]	RF energy harvester system to charge wearable devices	0.65	5.2	20	6.1	67
[63]	Textile rectenna for wearable power harvesting	1.2	0.82	−20	1	41.8
[163]	Sub-1 GHz wearable textile rectenna	1.8	0.915	10	3.2	38
[61]	Single-thread RFEH wearable rectenna	——	0.915	0	1.8	55
[143]	Dual-polarized wearable rectenna	——	2.4	2	4.2	74
[164]	Fixable metamaterial-based RF energy harvester	——	5.8	12	0.00298	98

**Table 6 sensors-22-06371-t006:** Comparison of implantable RF energy harvester devices.

Reference	Implantable Device	Method	Test Model	Size	Frequency (GHz)	Depth	Transfer Distance	Output Power	PTE (%)
[200]	Wireless charging system for an implanted capsule robot	Near field	Human torso	Diameter: 1 cm Length: 2 cm	0.0003	——	——	1 W	10
[84]	A Dual-Band Implantable Rectenna	Near field	Minced Pork	16 × 14 × 1.27 mm	0.915	1	50 cm	25 μW	0.006
[201]	Implantable wireless optogenetic device	Near field	Mouse tissue	10 mm${}^{3}$	1.5	3	——	15.7 mW	0.5
[202]	Implantable loop antennas	Near field	Minced pork	20 × 10.5 mm	0.433	3	——	1 mW	0.1
[85]	WPT system for multipurpose biomedical implants	Mid field	Minced pork	9.4 mm${}^{3}$	1.47	5.5	60 mm	10.7 mW	1.07
[165]	Bioelectronic microdevices	Mid field	Porcine animal	12 mm${}^{3}$	1.6	4	4.5	0.45 mW	0.06
[191]	Broadband high-efficiency rectifier	Mid field	Porcine animal	16 × 11 mm${}^{2}$	1.5	5.5	55 mm	0.9 mW	0.56
[183]	Midfield WPT for Deep-Tissue Biomedical Implants	Mid field	Minced pork	93.6 mm${}^{3}$	1.5	4.5	55 mm	5.6 mW	0.56
[203]	Multichannel passive neurosensing system	Mid field	Pig Skin	1632 mm${}^{3}$	2.4	0.25	2.5 mm	0.6 mW	15
[194]	Implantable rectenna	Far field	Minced pork	4 × 8 mm	2.45	0.3	50 mm	1 W	0.007
[195]	Implantable miniaturized optoelectronic systems	Far field	Mouse tissue	2 mm	2.34	0.5	——	0.1 mW	——
[65]	Epidermal RF power harvester	Far field	Skin	2160 mm${}^{2}$	1	——	1.5 m	32 mW	0.2

## Data Availability

Not applicable.

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
