# Peer review of "Self-Sustainable Biomedical Devices Powered by RF Energy: A Review"

_sensors, 2022, doi:10.3390/s22176371_

Round 1
Reviewer 1 Report
Dear authors;
Your paper entitled “Self-Sustainable Biomedical Devices Powered by RF Energy: A Review” I have several comments that I wish to be useful for you:
1- The abstract needs more interest and rewriting some paragraphs.
2- There are still some aspects that can be improved (for grammar and punctuations). Improve the technical writing of your paper, where there are several grammatical errors and spelling I think they need to be checked out.
3- The conclusion needs more efforts to elaborate the achieved results with respect to the future work,
4- The practical part is very important,
5- Future work is an important part of the conclusion.
I loved this work and I feel it is very good. I hope these comments would help you improve this work after a major revision.
Regards
Author Response
Response to Reviewer 1 Comments
We thank the reviewer for carefully reviewing our paper and giving us an opportunity to revise it. The authors highly appreciate the reviewer's words about the article. We have carefully addressed all the comments by the reviewer. The revised parts are highlighted in blue in the resubmitted manuscript. The responses to the reviewer's comments are listed as follows:
Point 1: The abstract needs more interest and rewriting some paragraphs.
Response 1: We thank the reviewer for pointing this out; the abstract is modified precisely to reflect the main contribution of the review article.
Point 2: There are still some aspects that can be improved (for grammar and punctuations). Improve the technical writing of your paper, where there are several grammatical errors and spelling I think they need to be checked out.
Response 2: We sincerely appreciate the reviewer’s point; the technical writing aspects are improved, and we have corrected the grammatical and spelling errors.
Point 3: The conclusion needs more efforts to elaborate the achieved results with respect to the future work.
Response 3: We thank the reviewer for the suggestion; the conclusion is revised to elaborate the paper's findings accurately and indicates future research directions.
Point 4: The practical part is very important.
Response 4: We thank the reviewer for the concern; the applications of the RFEH in wearable and implantable medical devices are discussed substantially, and we added more references to the practical part with the corresponding figures in the revised manuscript.
Point 5: Future work is an important part of the conclusion.
Response 5: We thank the reviewer for the suggestion; the conclusion is revised to elaborate the future work for RFEH in wearable and implantable medical devices. In addition, we discussed the future research directions in the discussion parts.

Reviewer 2 Report
This manuscript presents an interesting review of the radio frequency energy harvesting (RFEH) technique to power wearable and implantable medical devices. In addition, this review includes the integration of antenna, matching network, and rectifier topologies. Also, this manuscript reports the advantages and bottlenecks of the RFEH technology applied to wearable and implantable medical devices. Potential applications of RFEH technology in wearable and implantable medical devices are indicated. This review can be improved considering the following comments:
1.-Introduction should add the main advantages of the RFEH technology compared with other nanogenerator types.
2.-This manuscript should include schematics or figures of RFEH devices and their applications.
3.-The quality of Figure 5 should be improved.
4.- This review should add more discussions on the materials, biocompatibility, stability, and durability of RFEH devices.
5.-This manuscript should include more discussion on the main limitations or challenges of the RFEH technology to be applied to wearable and implantable medical devices.
Author Response
Response to Reviewer 2 Comments
We thank the reviewer for carefully reviewing our paper and giving us an opportunity to revise it. We have carefully addressed all the comments by the reviewer. Following his suggestions, we have revised the paper substantially. The revised parts are highlighted in blue in the resubmitted manuscript. The responses to the reviewer's comments are listed as follows:
Point 1: Introduction should add the main advantages of the RFEH technology compared with other nanogenerator types.
Response 1: We thank the reviewer for the observation; the main advantages of the RFEH technology over the other energy harvesting technologies are added to the introduction.
Point 2: This manuscript should include schematics or figures of RFEH devices and their applications.
Response 2: We thank the reviewer for the suggestion; more works about the RFEH applications in wearable and implantable medical devices are added with the corresponding figures.
Point 3: The quality of Figure 5 should be improved.
Response 3: Thank you for pointing this out; we have redrawn figure 5 and inserted a high-resolution photo in the revised manuscript.
Point 4: This review should add more discussions on the materials, biocompatibility, stability, and durability of RFEH devices.
Response 4: Thank you for the suggestion; more discussions on the biocompatible materials are added and highlighted in the implantable antenna design part and the discussion part of the implantable medical devices. The performance and design aspects of the RFEH system for wearable and implantable devices are analyzed extensively in the revised manuscript.
Point 5: This manuscript should include more discussion on the main limitations or challenges of the RFEH technology to be applied to wearable and implantable medical devices.
Response 5: Thank you for the recommendation; More discussion on the main challenges of applying the RFEH technology in wearable and implantable medical devices are included in the discussion parts of the revised manuscript.

Round 2
Reviewer 2 Report
The authors have improved their manuscript considering the reviewer's comments. This revised manuscript can be accepted for publication in Sensors.